# Influences of Initial Empiric Antibiotics with Ampicillin plus Cefotaxime on the Outcomes of Neonates with Respiratory Failure: A Propensity Score Matched Analysis

**DOI:** 10.3390/antibiotics12030445

**Published:** 2023-02-23

**Authors:** Mei-Chen Ou-Yang, Jen-Fu Hsu, Shih-Ming Chu, Ching-Min Chang, Chih-Chen Chen, Hsuan-Rong Huang, Peng-Hong Yang, Ren-Huei Fu, Ming-Horng Tsai

**Affiliations:** 1School of Medicine, College of Medicine, Chang Gung University, Taoyuan 333, Taiwan; 2Division of Neonatology, Department of Pediatrics, Chang Gung Memorial Hospital, Kaohsiung 833, Taiwan; 3Division of Neonatology, Department of Pediatrics, Linkou Chang Gung Memorial Hospital, Taoyuan 333, Taiwan; 4Division of Pediatric Gastrointestinal Disease, Department of Pediatrics, Chang Gung Memorial Hospital, Chiayi 613, Taiwan; 5Division of Neonatology and Pediatric Hematology/Oncology, Department of Pediatrics, Chang Gung Memorial Hospital, Yunlin 638, Taiwan

**Keywords:** empiric antibiotics, intubation, neonates, MDR bacteremia, mortality

## Abstract

**Background:** Empiric antibiotics are often prescribed in critically ill and preterm neonates at birth until sepsis can be ruled out. Although the current guideline suggests narrow-spectrum antibiotics, an upgrade in antibiotics is common in the neonatal intensive care unit. The impacts of initial broad-spectrum antibiotics on the outcomes of critically ill neonates with respiratory failure requiring mechanical intubation have not been well studied. **Methods:** A total of 1162 neonates from a tertiary level neonatal intensive care unit (NICU) in Taiwan who were on mechanical ventilation for respiratory distress/failure at birth were enrolled, and neonates receiving ampicillin plus cefotaxime were compared with those receiving ampicillin plus gentamicin. Propensity score-matched analysis was used to investigate the effects of ampicillin plus cefotaxime on the outcomes of critically ill neonates. **Results:** Ampicillin plus cefotaxime was more frequently prescribed for intubated neonates with lower birth weight, higher severity of illness, and those with a high risk of early-onset sepsis. Only 11.1% of these neonates had blood culture-confirmed early-onset sepsis and/or congenital pneumonia. The use of ampicillin plus cefotaxime did not significantly contribute to improved outcomes among neonates with early-onset sepsis. After propensity score-matched analyses, the critically ill neonates receiving ampicillin plus cefotaxime had significantly worse outcomes than those receiving ampicillin plus gentamicin, including a higher risk of late-onset sepsis caused by multidrug-resistant pathogens (11.2% versus 7.1%, *p* = 0.027), longer duration of hospitalization (median [IQR], 86.5 [47–118.8] days versus 78 [45.0–106.0] days, *p* = 0.002), and a significantly higher risk of in-hospital mortality (14.2% versus 9.6%, *p* = 0.023). **Conclusions:** Ampicillin plus cefotaxime should not be routinely prescribed as the empiric antibiotics for critically ill neonates at birth because they were associated with a higher risk of infections caused by multidrug-resistant pathogens and final worse outcomes.

## 1. Introduction

Initiation of empiric antibiotics at birth is a common practice in very low birth weight (LVBW, BW < 1500 g) and critically ill neonates in the neonatal intensive care unit (NICU) [1,2]. In critically ill neonates requiring mechanical ventilation due to respiratory distress at birth, empiric antibiotics are indicated for possible congenital pneumonia and/or early-onset sepsis [3,4]. The use of initial empiric antibiotics for these high-risk neonates is recommended by the current American Academy of Pediatrics (AAP) guidelines and should be stopped within three days after bacterial sepsis can be ruled out [5]. Prolonged use of empiric antibiotics for more than 5 to 7 days is associated with adverse outcomes, including an increased risk of subsequent late-onset sepsis, necrotizing enterocolitis, and final mortality [4,6,7].

The use of broad-spectrum antibiotics, such as ampicillin plus a third-generation cephalosporin, is known to disturb the gut microbiota and increase the risk of the emergence of antibiotic-resistant pathogens and invasive candidiasis [8,9,10,11]. Despite this, the use of empiric broad-spectrum antibiotics is common among neonates born with risk factors such as prolonged rupture of membranes and maternal fever, as well as neonates with clinical signs of sepsis and elevated biomarkers of infection [1,2,12]. A significant proportion of neonates receive broad-spectrum antibiotics at birth despite negative blood cultures because inadequate antibiotic treatment in patients with early-onset sepsis will increase the risk of mortality [2,11,13]. Most previous studies enrolled all NICU neonates for analyses, whereas few studies have focused on neonates with high severity of illness and/or severe respiratory failure [2,3,4,14,15,16]. In this study, we aimed to compare the outcomes of critically ill neonates who were treated empirically with ampicillin plus cefotaxime at birth to those treated with ampicillin plus gentamicin.

## 2. Results

During the study period, a total of 1221 neonates with respiratory failure requiring intubation were identified. After excluding neonates with severe congenital anomalies (n = 12), neonates who received ampicillin plus gentamicin or cefotaxime plus metronidazole (n = 14), and neonates who died within the first 48 h of life (n = 33), a total of 1162 neonates were enrolled for analyses. Among them, 670 neonates (57.7%) received ampicillin plus cefotaxime as initial empiric antibiotics, and 492 neonates (42.3%) received ampicillin plus gentamicin as initial empiric antibiotics. The patients’ demographics, perinatal issues, underlying clinical diagnoses of respiratory failure, laboratory results, and ventilator parameters are presented in Table 1. In our cohort, more than half (55.9%) of our patients were extremely preterm neonates (gestational age (GA) < 28 weeks), and 56.5% were extremely low birth weight infants (BBW < 1000 g). The majority of our cohort had mechanical intubation soon after birth, and only 17.6% (n = 204) had intubation after 12 h of life. A total of 353 neonates (30.4%) had premature rupture of the membrane, and 154 neonates (13.3%) had a maternal history suggestive of infection, including maternal fever and chorioamnionitis.

In our cohort, only a total of 54 episodes of EOS were documented during the study period. *E. coli* (n = 22, 40.7%) was the most common pathogen followed by Group B *Streptococcus* (n = 13, 24.1%) and *Listeria monocytogenes* (n = 4, 7.4%). Although Gram-negative bacteria accounted for 51.9% (n = 28) of all EOS episodes, only three EOS episodes were caused by MDR pathogens. A total of six patients died of EOS sepsis and associated infectious complications, and only two of them were treated with initial inappropriate antibiotics. Except for the 129 neonates who received therapeutic antibiotics for ≥7 days due to documented EOS, clinically suspected sepsis, and/or congenital pneumonia, all other cases had empiric antibiotics stopped at 72 h of life in the cohort.

We found that neonates receiving ampicillin plus cefotaxime had significantly lower birth weight and were more preterm, had more perinatal insults, and had significantly higher severity of illness than the control group, including higher NTISS scores, more abnormal laboratory findings, more requirement of blood transfusions and inotropic agents use, and higher rates of EOS and/or congenital pneumonia. Most of these neonates received initial empiric antibiotics for only 3 days, and only 11.1% (n = 129) had received empiric antibiotics for more than 5 days. Neonates receiving ampicillin plus cefotaxime were significantly more likely to receive empiric antibiotics ≥five days than the control group (33.4% vs. 8.9%, *p* = 0.014).

### 2.1. Hospital Course and Outcomes of Neonates Treated with Ampicillin plus Cefotaxime Compared to Those Treated with Ampicillin plus Gentamicin

For outcome analyses (Table 2), we found that neonates receiving ampicillin plus cefotaxime had significantly longer durations of intubation, use of mechanical ventilators, use of total parenteral nutrition, and hospitalization. During hospitalization, neonates receiving ampicillin plus cefotaxime had a significantly higher rate of late-onset sepsis and multidrug-resistant bacteremia (*p* < 0.001 and *p* = 0.015, respectively) than the control group, although the rate of experiencing any nosocomial infections was comparable between these two groups. The in-hospital mortality rate was also significantly higher in neonates receiving ampicillin plus cefotaxime than the controls (14.2% vs. 9.6%, *p* = 0.018).

### 2.2. The Propensity Score-Matched Analyses

Because neonates receiving ampicillin plus cefotaxime were significantly more preterm and had lower birth weights than the controls, a propensity score-matching analysis was conducted to evaluate the effects of different empiric antibiotics (Table 3). After propensity matching for birth weight, gestational age, the Apgar score at the 5th minute, and the first NTISS scores soon after birth, 492 pairs were matched. We found that preterm neonates receiving ampicillin plus cefotaxime had significantly worse outcomes than those receiving ampicillin plus gentamicin, including a higher risk of late-onset sepsis caused by MDR pathogens (11.2% versus 7.1%, *p* = 0.027), longer duration of hospitalization (median [IQR], 86.5 [47–118.8] days versus 78 [45.0–106.0] days, *p* = 0.002), and a significantly higher risk of final in-hospital mortality (14.2% versus 9.6%, *p* = 0.023). The Kaplan–Meier graph also shows a higher risk of final in-hospital mortality in neonates receiving ampicillin plus cefotaxime (*p* = 0.027 by log-rank test) (Figure 1) when compared with those receiving ampicillin plus gentamicin.

## 3. Discussion

Recent systemic reviews and meta-analyses found that there is currently inadequate evidence to support any empiric antibiotic regimen being superior to another for neonates with suspected EOS [17]. For empiric antibiotic use at birth, recent studies have advanced to suggest that low-risk infants, defined as preterm infants delivered by cesarean section due to maternal noninfectious diseases in the absence of labor induction, rupture of membranes or intrapartum antibiotic prophylaxis, may be managed without evaluation for EOS and routine use of empiric antibiotics at birth [1,14,15,16,18]. The latest guideline for neonates with respiratory distress syndrome recommends that the shortest possible course of empirical antibiotics should be 36 h after screening [19]. However, in our clinical practice, due to fear of clinical deterioration, an upgrade of empiric antibiotics was often decided without documented evidence of EOS. We found that ampicillin plus cefotaxime did not significantly improve the outcomes of neonates with respiratory failure at birth. Instead, this combination regimen contributed to an increased risk of MDR bacteremia, prolonged duration of hospitalization, and a higher risk of in-hospital mortality after adjustment for covariate factors.

In our institute, ampicillin plus gentamicin is usually prescribed as the initial empiric antibiotics for common premature or less ill neonates at birth because this regimen can provide adequate coverage of neonatal sepsis [20]. However, for neonates with respiratory failure requiring intubation and/or those with higher illness severity at birth, ampicillin plus cefotaxime is more frequently prescribed. This policy is based on clinicians’ judgement and the AAP guidelines [5,21], which support the use of broad-spectrum antibiotics in neonates with a high risk of severe sepsis, meningitis, sepsis-associated organ dysfunction, or contraindication of gentamicin. Although EOS is more likely to occur in term-born or late-preterm neonates [22], preterm infants have a higher incidence of clinical sepsis and sepsis-attributable mortality rate than term infants [22,23]. Another concern is the renal toxicity of gentamicin, which is suggested to be used causally in extremely preterm neonates with suboptimal renal function during the most critically ill period [24].

Previous studies concluded that neonates using ampicillin plus cefotaxime had a significantly higher mortality rate than those receiving ampicillin plus gentamicin [21]. It is inevitable that the bias existed in both the Clark et al. study [21] and this study because of the retrospective study design and because clinicians tended to upgrade empiric antibiotics in neonates with higher illness severity. Therefore, a propensity score matching analysis was applied to overcome this limitation. Additionally, almost all neonates with empiric antibiotics were enrolled in the previous studies [21,25] and the real effects of broad-spectrum antibiotics would be masked by the relatively stable neonates, who were much less likely to have LOS or comorbidities. After the propensity score analysis, we found that ampicillin plus cefotaxime increased the risk of final adverse outcomes. The higher risk of final in-hospital mortality may be associated with the greater likelihood of MDR bacteremia, which has been documented to be associated with a higher risk of treatment failure and infectious complications [10,26,27,28]. The occurrence of MDR bacteremia and infectious complications will lead to a higher risk of recurrent sepsis and then a vicious circle will occur, which can explain the prolonged duration of hospitalization in neonates with initial broad-spectrum antibiotics [27,29,30].

Based on our data and the literature, culture-confirmed early-onset sepsis occurs in only 1.8–3% of VLBW infants [1,22,24] and only 4.6% of our subjects who presented with respiratory failure at birth. We also found that the MDR pathogens accounted for only 2.4% of all EOS episodes. Among neonates with EOS and meningitis, most were complicated *Streptococcus agalactiae* infections. Therefore, we can conclude that ampicillin plus cefotaxime did not contribute significantly to improving the outcomes, even among neonates with documented EOS in our cohort.

Recent evidence supports the concept that broad-spectrum antibiotics could destroy gut microbiota, especially those of extremely preterm neonates during the important developmental stage of the early postpartum period [31,32]. In our cohort, we routinely used oral probiotics Infloran capsules (Desma Health care, Chiasso, Switzerland) containing *Lactobacillus acidophilus* and *Bifidobacterium bifidum*, a capsule (250 mg) once daily for extremely preterm neonates (GA ≤ 28 weeks) in recent years. The incidence rate of late-onset sepsis did not change significantly when compared with our previous study from ten years ago [26]. The beneficial effects of probiotics on reducing late-onset sepsis remain debatable, although most studies have concluded that probiotics can decrease the incidence rate of necrotizing enterocolitis [32,33].

There were some limitations in this study. This was a retrospective cohort study from a single center and the choice of antibiotics depended on the attending physicians. The outcomes of critically ill neonates in the NICU are affected by various characteristics, such as extremely preterm, VLBW, and other chronic comorbidities. Therefore, the conclusion of this study may be affected by many biases and less applicable to other institutes and other countries. Another bias was the non-standard treatment and shift of empiric antibiotics, although only 4.3% (n = 29) of the study group had a modification of empiric antibiotics. The modifications of empiric antibiotics during the critically ill period were affected by the bias of higher severity of illness. Although propensity score matching was performed in this study, it is inevitable that some potential factors or unidentified variables that might affect outcomes were missed. It is not possible to conduct a randomized control trial to investigate the significant impacts of different empirical antibiotics on the outcomes. However, strict antibiotic stewardship programs can be considered for early-onset sepsis and avoidance of broad-spectrum antibiotics should be considered.

## 4. Methods

### 4.1. Study Design, Setting, and Patients

We conducted a cohort study and retrospectively reviewed all neonates with respiratory failure requiring mechanical ventilation within the first day of life who were admitted to the NICUs of Chang Gung Memorial Hospital (CGMH) between January 2015 and December 2020. The NICUs of CGMH contained a total of 3 units and a total capacity of 47 beds equipped with ventilators and 60 beds of special care nurseries. The annual number of inpatients in these NICUs is 800, including more than one-fourth of them who were transferred from other hospitals. In our institute, criteria for mechanical ventilation include the following: if we fail to maintain PaO_2_ > 60, a pH value > 7.25 and the requirement of fraction of inspired oxygenation (FiO_2_) > 60 using a noninvasive ventilator, or the presence of hypercarbia (PaCO_2_ > 60) or persistent apnea despite using a noninvasive ventilator. Neonates with severe congenital anomalies and those who died within 48 h of life were excluded from the analyses. This study was approved by the institutional review board of CGMH, and all methods in this study were performed in accordance with the relevant guidelines and regulations. The need for informed consent was waived because all patient records/information were anonymized and deidentified prior to analysis.

In our institute, all neonates on mechanical ventilation for respiratory failure or respiratory distress would receive empiric antibiotics with either ampicillin plus gentamicin or ampicillin plus cefotaxime, depending on the decisions of the attending physicians. The empiric antibiotics were administered soon after septic work up and a blood culture from peripheral vein was taken. Only cases with positive blood culture-confirmed early-onset sepsis (EOS, onset of sepsis within the first seven days of life), clinically suspected sepsis, or congenital pneumonia confirmed by the attending physicians will be administered with therapeutic antibiotics for seven or more than seven days, depending on therapeutic responses. Otherwise, the initial empiric antibiotics were ceased three days later.

### 4.2. Data Collection and Definition

Patient demographics, their perinatal history, use of empiric antibiotics, EOS, and treatment outcomes of initial respiratory failure or distress were retrospectively reviewed for all neonates who were enrolled in the study. For neonates with documented EOS or congenital pneumonia, therapeutic antibiotics will usually be modified according to the culture results. For clinically suspected sepsis, decisions to modify antibiotics were made on clinical judgement based on therapeutic responses. The severity of illness was routinely evaluated on the first day of life using the neonatal therapeutic intervention scoring system (NTISS) [34]. All patients’ hospital courses were reviewed and collected until discharge or death. The courses of hospitalization, including nosocomial infections, late-onset sepsis (LOS, onset of sepsis after the 7th day of life), and duration of ventilation and hospitalization were also recorded.

The definitions of EOS, LOS, nosocomial infections, and congenital pneumonia were based on the updated criteria of the Centers for Disease Control and Prevention (CDC) [35,36]. All comorbidities of prematurity, including intraventricular hemorrhage, respiratory distress syndrome, neurological sequelae, bronchopulmonary dysplasia (BPD), necrotizing enterocolitis, short bowel syndrome, and periventricular leukomalacia were based on the latest updated diagnostic criteria in the standard textbook of neonatology [37].

### 4.3. Statistical Analysis

Parametric variables are expressed as mean (standard deviation, SD), and continuous variables with non-parametric distributions are expressed as median (interquartile range, IQR). Comparisons between continuous variables of two subgroups were analyzed using paired Student’s *t*-test and the paired Wilcoxon rank sum tests. Categorical variables were compared using Chi-square tests or Fisher’s exact tests. All *p* values were two-tailed, and *p* values < 0.05 were considered to be statistically significant. All statistical analyses were performed using SPSS (version 21.0; IBM, Armonk, NY, USA).

We categorized these critically ill neonates with mechanical ventilation into two groups: neonates using ampicillin plus cefotaxime (the study group) and those using ampicillin plus gentamicin (the control group). Neonates initially using ampicillin plus gentamicin but later upgraded to ampicillin plus cefotaxime were categorized as the study group. The primary outcome was final in-hospital mortality. The duration of hospitalization, the incidence rate of LOS, and the occurrence of multidrug-resistant (MDR) pathogens bacteremia, defined based on our previous study [10], were also compared.

Propensity scores using multivariable logistic regression models were calculated within the two groups, which included covariates that may affect the likelihood of patients to receive ampicillin plus either cefotaxime or gentamicin. The outcomes of interest were unbalanced between the two groups before matching. Birth weight and/or gestational age, the NTISS scores soon after birth, which indicated the initial severity of illness, and some risk factors of early-onset sepsis were analyzed in the multivariable logistic regression [12,16]. Matching based on propensity scores incorporating different sets of covariates was performed using a 1:1 nearest-neighbor algorithm. We identified the best-matched cohort based on the most balanced distribution of propensity scores and the best balance in individual covariates between the study and control groups.

## 5. Conclusions

In conclusion, we found that ampicillin plus cefotaxime neither significantly improved the outcomes of neonates with respiratory failure requiring intubation at birth nor provided more coverage for neonatal early-onset sepsis than another combined regimen of ampicillin plus gentamicin. We found that most of the pathogens of EOS were susceptible to ampicillin plus gentamicin, and relatively few patients died of EOS and/or congenital pneumonia. Ampicillin plus cefotaxime should be casually prescribed only in selected critically ill neonates because they were associated with a higher risk of infections caused by MDR pathogens and final worse outcomes. Additionally, future studies that optimize early antibiotic prescription among critically ill and preterm infants without adverse effects are warranted in the future.

## Figures and Tables

**Figure 1 antibiotics-12-00445-f001:**
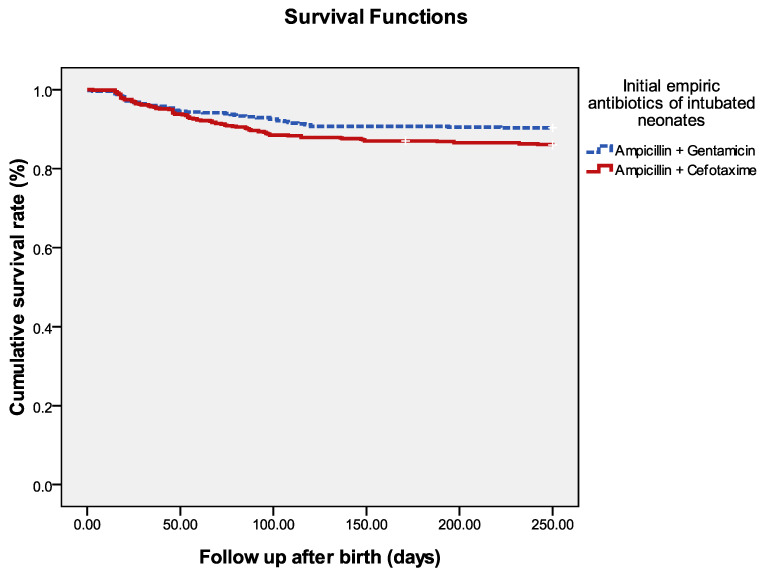
Survival of all neonates with mechanical intubation at birth. The Kaplan–Meier graph is stratified by neonates receiving ampicillin plus cefotaxime and those receiving ampicillin plus gentamicin. The log-rank test = 0.027.

**Table 1 antibiotics-12-00445-t001:** Patient demographics, characteristics, and clinical presentation of all neonates with respiratory failure.

Characteristics	All Study Subjects (Total n = 1162)	Neonates Using Ampicillin + Cefotaxime (Total n = 670)	Neonates Using Ampicillin + Gentamicin (Total n = 492)	*p* Values
Case demographics				
Gestational age (weeks), median (IQR)	27.3 (25.3–31.0)	26.5 (25.0–30.3)	28.0 (26.0–31.8)	<0.001
Birth weight (g), median (IQR)	940.0 (740.0–1480.0)	877.0 (700.0–1331.0)	1040.0 (800.0–1645.0)	<0.001
Gender (male/female), n (%)	700 (60.2)/462 (39.8)	405 (60.4)/265 (39.6)	295 (60.0)/197 (40.0)	0.903
Birth by NSD/cesarean section, n (%)	366 (31.5)/796 (68.5)	242 (36.1)/428 (63.9)	124 (25.2)/368 (74.8)	<0.001
5 min Apgar score < 7, n (%)	346 (29.8)	224 (33.4)	122 (24.8)	0.001
Inborn/outborn, n (%)	940 (80.9)/222 (19.1)	536 (80.0)/134 (20.0)	404 (82.1)/88 (17.9)	0.406
Premature rupture of membrane, n (%)	353 (30.4)	250 (37.3)	103 (20.9)	<0.001
Maternal fever, n (%)	149 (12.8)	108 (16.1)	41 (8.3)	<0.001
Intrapartum antibiotic prophylaxis, n	86 (7.4)	40 (6.0)	46 (9.3)	0.032
Chorioamnionitis, n (%)	17 (1.5)	6 (0.9)	11 (2.2)	0.082
Perinatal asphyxia, n (%)	213 (18.3)	142 (21.2)	71 (14.4)	0.004
Diagnoses of respiratory failure, n (%)				
Respiratory distress syndrome (≥Gr II)	779 (67.0)	452 (67.5)	327 (66.5)	0.752
Transient tachypnea of newborn	56 (4.8)	25 (3.7)	31 (6.3)	0.052
Complicated cardiovascular diseases	19 (1.6)	9 (1.3)	10 (2.0)	0.243
Symptomatic patent ductus arteriosus	506 (43.5)	282 (42.1)	224 (45.5)	0.255
Persistent pulmonary hypertension of newborn	168 (14.5)	98 (14.6)	70 (14.2)	0.866
Pulmonary hemorrhage	63 (5.4)	39 (5.8)	24 (4.9)	0.515
Congenital diaphragmatic hernia	16 (1.4)	7 (1.0)	9 (1.8)	0.311
Air leak syndrome ^&^	107 (9.2)	62 (9.3)	45 (9.1)	1.000
Meconium aspiration syndrome	32 (2.8)	18 (2.7)	14 (2.8)	0.858
Sepsis and/or congenital pneumonia	129 (11.1)	95 (14.2)	34 (6.9)	<0.001
Hydrops fetalis	32 (2.8)	15 (2.2)	17 (3.4)	0.182
Initial ventilator requirement *, n (%)				0.560
Intubation with mechanical ventilation	842 (72.5)	482 (71.9)	360 (73.2)	
Initial FiO_2_ ≤ 50	507 (43.6)	299 (44.6)	208 (42.3)	
Initial FiO_2_ > 50	335 (28.8)	183 (27.3)	152 (30.9)	
On high frequency oscillatory ventilation	320 (27.5)	188 (28.1)	132 (26.8)	
High setting (FiO_2_ ≤ 50)	139 (12.0)	84 (12.5)	55 (11.2)	
Low setting (FiO_2_ > 50)	181 (15.6)	104 (15.5)	77 (15.7)	
Oxygenation index, median (IQR)	10.0 (6.0–18.0)	10.0 (5.0–18.0)	9.0 (5.0–15.0)	0.050
AaDO_2_, median (IQR)	248.5 (156.0–441.0)	249.0 (160.0–450.0)	238.5 (146.5–417.8)	0.226
Use of iNO	182 (15.7)	111 (16.6)	71 (14.4)	0.329
Clinical features *, n (%)				
Intravascular volume expansion	948 (81.6)	578 (86.3)	370 (75.2)	<0.001
Requirement of cardiac inotropic agents	809 (69.6)	494 (73.7)	315 (64.0)	<0.001
Metabolic acidosis	409 (35.2)	255 (38.1)	154 (31.3)	0.018
Coagulopathy	851 (73.2)	528 (78.8)	323 (65.7)	<0.001
Requirement of blood transfusion **	330 (28.4)	201 (30.0)	129 (26.2)	0.167
Laboratory data at birth				
Leukocytosis or leukopenia	288 (24.8)	205 (30.6)	83 (16.9)	<0.001
Shift to left in WBC (immature > 20%)	75 (6.5)	33 (4.9)	42 (8.5)	0.015
Anemia (hemoglobin level < 11.5 g/dL)	203 (17.5)	129 (19.3)	74 (15.0)	0.072
Thrombocytopenia (platelet < 150,000/uL)	238 (20.5)	132 (19.7)	106 (21.5)	0.462
C-reactive protein (mg/dL), median (IQR)	5.0 (2.0–20.5)	6.0 (2.0–24.0)	3.5 (1.5–10.5)	<0.001
Severity score at birth				
NTISS (median (IQR))	22.0 (20.0–26.0)	22.5 (20.0–26.0)	22.0 (19.0–25.0)	<0.001

FiO_2_: fraction of inspired oxygen; NSD: normal spontaneous delivery; IQR: interquartile range; iNO: inhaled nitric oxide; HFOV: high-frequency oscillatory ventilator; WBC: white blood cell; NTISS score: Neonatal Therapeutic Intervention Scoring System; Leukocytosis: white blood cell count >20,000/L; Leukopenia: white blood cell count < 4000/L. ^&^ Including pneumothorax, pneumomediastinum, and pulmonary interstitial emphysema.* At onset of respiratory failure.** Including leukocyte-poor red blood cell and/or platelet transfusion.

**Table 2 antibiotics-12-00445-t002:** Hospital courses and outcomes of neonates using ampicillin plus cefotaxime compared with those using ampicillin plus gentamicin as empiric antibiotics.

Characteristics	All Study Subjects (Total n = 1162)	Neonates Using Ampicillin + Cefotaxime (Total n = 670)	Neonates Using Ampicillin + Gentamicin (Total n = 492)	*p* Values
Early-onset sepsis, n (%)	54 (4.6)	33 (4.9)	21 (4.3)	0.688
Late-onset sepsis during hospitalization, n (%)	380 (32.7)	250 (37.3)	130 (26.4)	<0.001
Multidrug resistant bacteremia *, n (%)	112 (9.6)	77 (11.5)	35 (7.1)	0.015
Neonates with any nosocomial infections ^#^, n (%)	539 (46.4)	302 (45.1)	237 (48.2)	0.312
Necrotizing enterocolitis (≥stage IIa)	14 (1.2)	9 (1.3)	5 (1.0)	0.787
Duration of TPN and/or intrafat (days), median (IQR)	38.0 (20.0–66.0)	42.0 (21.0–68.0)	32.0 (18.0–58.8)	<0.001
Duration of intubation (days), median (IQR)	11.5 (8.0–31.0)	13.0 (9.0–38.0)	10.0 (8.0–23.0)	<0.001
Duration of mechanical ventilation, median (IQR)	55.0 (28.0–80.0)	57.0 (29.0–80.0)	53.0 (27.8–69.0)	<0.001
Duration of hospitalization, median (IQR)	85.0 (48.0–114.0)	92.0 (49.8–124.3)	78.0 (45.0–106.0)	<0.001
Final in-hospital mortality, n (%)	142 (12.2)	95 (14.2)	47 (9.6)	0.018

* Including methicillin-resistant *Staphylococcus aureus* and multidrug-resistant Gram-negative pathogens, defined based on our previous study [10]. ^#^ Including late-onset sepsis, ventilator-associated pneumonia, catheter-related bloodstream infections, and clinical sepsis.

**Table 3 antibiotics-12-00445-t003:** Outcome comparisons of the two groups receiving different empiric antibiotics using the propensity score-matched analyses.

	Neonates Using Ampicillin + Cefotaxime (Total n = 492)	Neonates Using Ampicillin + Gentamicin (Total n = 492)	*p* Values
Propensity scores matching			
Gestational age (weeks), mean ± SD	29 ± 4.8	29 ± 4.3	0.295
Birth weight (g), median (IQR)	918.0 (733.5–1562.8)	1040 (800–1645.0)	0.410
5 min Apgar score, mean ± SD	7.2 ± 1.9	7.3 ± 1.9	0.294
NTISS scores at admission to NICU	22.0 (19.5–25.5)	22.0 (19.0–25.0)	0.188
Outcomes			
Late-onset sepsis, n (%)	152 (30.9)	130 (26.4)	0.139
Multidrug resistant bacteremia, n (%)	55 (11.2)	35 (7.1)	0.027
Neonates with any nosocomial infections, n (%)	220 (44.7)	237 (48.2)	0.306
Necrotizing enterocolitis (≥stage IIa), n (%)	7 (1.4)	5 (1.0)	0.584
Duration of TPN and/or intrafat (days), median (IQR)	37.0 (19.0–65.8)	32.0 (18.0–58.8)	0.071
Duration of intubation (days), median (IQR)	12.0 (9.0–38.0)	10.0 (8.0–23.0)	0.068
Duration of mechanical ventilation, median (IQR)	54.0 (28.0–78.0)	53.0 (27.8–69.0)	0.467
Duration of hospitalization, median (IQR)	86.5 (47–118.8)	78.0 (45.0–106.0)	0.002
Final in-hospital mortality, n (%)	70 (14.2)	47 (9.6)	0.023

NTISS: Neonatal Therapeutic Intervention Scoring System; IQR: interquartile range; NICU: neonatal intensive care unit.

## Data Availability

The datasets used/or analyzed during the current study are available from the corresponding author on reasonable request.

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
