# Peer review of "Influences of Initial Empiric Antibiotics with Ampicillin plus Cefotaxime on the Outcomes of Neonates with Respiratory Failure: A Propensity Score Matched Analysis"

_antibiotics, 2023, doi:10.3390/antibiotics12030445_

Round 1

Reviewer 1 Report

Thanks for performing this study. Of course, it has its limitation as you have mentioned because of being a retrospective study. I think as your two groups of study have different characteristics which have definitely affected the outcome of treatment, it's better if you don't compare the outcomes of two group of antibiotics before making two matching groups to be similar in everything else but empiric antibiotics. Also, maybe add. couple of more sentences on how different characteristics like being preterm, VLBW and so can affect the outcome of treatment in newborn with respiratory diseases or sepsis...I have added some comments and/or questions in the notes attached to the PDF file. Hope they are helpful. 

Good luck! 

Author Response

Dear reviewer:

     I appreciate your review and comments. Please see the attachment, thank you.

Best regard,

Tsai Ming Horng

Reviewer 2 Report

Dear authors,

This is an important study about the empiric antibiotics prescribed in critically ill and preterm neonates at birth and the impacts of initial broad spectrum antibiotics on the outcomes of this critically ill neonates with respiratory failure. Because of the retrospective nature of the study and due to the fact that the newborns received non-standardized antibiotic therapy, depending on the decision of the attending physician, many biases may appear. In any case, I think that the study is important enough to be published, after clarifying the following aspects:

-line 92-93 - what was the reason why the neonates initially using ampicillin plus gentamicin were later upgraded to ampicillin plus cefotaxime and why they were categorized as the study group? After how many days was this upgrade made and do you not think that this is an additional bias that could be eliminated by removing these patients from the study?

-line 138-139 - please reformulate this phrase, it is ambiguous

- line 144-145 - The same as above (please rephrase).

Author Response

(The authors gave the same response as above.)

Round 2

Reviewer 2 Report

Dear authors,

Thank you for your clarification and responses.

I still think that there are many biases regarding this study and all of these must appear in a specific section (limitation of the study). I suggest you to add in this special section a phrase regarding the non-standardized treatment and the shift in the empirical antibiotic treatment for a number of patients (and how many patients were treated in this specific way?). 

Best regards,

Author Response

(The authors gave the same response as above.)

Round 3

Reviewer 2 Report

Dear authors,

Thank you for your reply. I think your manuscript is ready for publication.